# Knowledge of the Legal Issues of Anti-Doping Regulations: Examining the Gender-Specific Validity of the Novel Measurement Tool Used for Professional Athletes

Draginja Vuksanovic Stankovic [1], Antonela Sinkovic [2], Damir Sekulic [3,*], Mario Jelicic [3] and Jelena Rodek [3]

1   Faculty of Law, University of Montenegro, 81000 Podgorica, Montenegro
2   Faculty of Kinesiology, University of Zagreb, 10000 Zagreb, Croatia
3   Faculty of Kinesiology, University of Split, 21000 Split, Croatia
*   Correspondence: dado@kifst.hr

**Abstract:** In the present study, we aim to assess the reliability and gender-specific validity of an original questionnaire (Q-LADR) in evaluating the knowledge of legal anti-doping regulations and to examine the gender-specific associations between Q-LADR and potential doping behavior (PDB) in senior-level professional athletes. The participants were team-sport players from Croatia and Montenegro ($n = 479$, 179 females, $21.3 \pm 3.3$ years of age). Apart from Q-LADR, they were tested in sociodemographic, sport, and doping factors. The results show the proper test–retest reliability of the Q-LADR (Cohen's kappa = 0.65; average percentage of the equally responded questions: 84%). Men achieved higher scores for the Q-LADR than women ($t$-test = 9.55, $p < 0.001$). The Q-LADR score was correlated with age in men, and with number of doping tests and sport success in men and women. Lower Q-LADR scores were correlated with neutral (in women) and positive doping attitudes (in men and women). The results confirm the importance of testing knowledge on the legal issues of anti-doping regulations for athletes, with the possible applicability of findings in the global fight against doping in sport. In order to provide equal opportunities for all to be involved in professional sport, special attention should be paid to vulnerable groups (i.e., women, younger athletes, and those who have not achieved sport success).

**Keywords:** questionnaire; validity; legislative; sport law; integrity

## 1. Introduction

Doping, namely, the use of a substance or technique to illegally improve athletic performance, is one the most critical problems in modern sport. Apart from being a serious health-threatening behavior associated with numerous detrimental health-related problems (i.e., hypertension, diabetes, blood clotting, insomnia, and anxiety), doping is considered an unfair way of improving an athletes' capacities and, therefore, contradicts the main essence of fair play in sport [1,2]. Not surprisingly, there is a growing body of research examining (i) methods for the detection of doping in athlete specimens, (ii) the negative health consequences of doping, and (iii) factors associated with doping behavior in athletes [3–6]. From the perspective of sustainability of sport as a global phenomenon that should be accessible and affordable around the world, all three specified research topics deserve attention. However, it is clear that the information obtained for (i) methods of detection and (ii) health consequences should be considered as "universal", and the findings of the studies dealing with these issues are globally generalizable regardless of the country/region/sport of origin. Meanwhile, the knowledge concerning factors of influence on doping behavior is known to be sport- and community (socioculturally)-specific, and there is a growing body of evidence that this issue should be observed as being gender-specific [7–9].

The rationale for adopting a gender-specific approach in studying the association of specific factors with doping behavior is based on the following reasons. First, in order to achieve sport performance results, female and male athletes are differentially oriented toward doping, simply because of physiological and psychological gender differences [10]. Studies have mostly—but not always—found that female athletes have a lower tendency of doping than their male peers within the same sport [3,7,8,11]. This was supported by a recent French study providing evidence of female athletes being less prone to doping than their male peers, with a significantly lower usage of hormones (i.e., anabolic agents). However, females were more prone to using glucocorticoids, beta-2 agonists, and diuretics [10]. Finally, a gender-specific approach is directly confirmed in the previous studies when authors confirmed that males and females often differed in factors they individually considered as being important barriers against their personal doping behavior (e.g., factors of hesitation (FHs)).

For example, a study on college-level athletes identified the differential influence of various FHs in men and women, with familial education level being the important FH against doping behavior in women, and religiousness (religiosity) being the important FH in men [12]. Even when a specific issue was investigated (i.e., health hazard of doping behavior), the genders differed in the ranking of specific consequences that should be observed as an FH against doping. For example, a study conducted on young athletes involved in different sports evidenced health hazards related to doping usage as highly important FHs in both males and females, but within this group of FHs, females ranked possible body malformations (e.g., hirsutism and development of male pattern baldness) as the most important negative consequence of doping usage, while males ranked immune function problems as being the most important FH [9]. Finally, although the authors of the previously mentioned studies highlighted the possible influence of gender-specific popularity of a certain type of sport on the differentiation in FHs between males and females (i.e., some sports are considered as being typically "male sports", while some are considered as "female sports"), and while sports themselves frequently differ in tendency toward a specific type of doping), it seems that sport type should not be observed as an important covariate based on the previous findings regarding the FHs in male and female athletes. Namely, more recent studies examining specific sport and/or group of sports supported these previous results and identified clear gender-specific FHs for athletes involved in the same sport [3,8,13].

Despite the global interest regarding doping in sport, there is an evident lack of knowledge about the legal issues (i.e., law regulations and legal regulations) and their eventual influence on doping behavior in athletes [14–16]. This is particularly interesting given that legal regulations are recognized as one of the most important mechanisms in the global fight against doping and is generally defined in the Law on Sports in most of the countries that are members of the International Olympic Committee [17–19]. However, there is and evident lack of: (i) specific measurement tools examining knowledge about the legal regulations of the anti-doping policy, and (ii) knowledge of the eventual association between this type of knowledge and the doping behavior of athletes. This is particularly important if we know that the global anti-doping authority, the World Anti-Doping Agency (WADA), adopted the World Anti-Doping Code (WADC) and established it as the most important by-law that contains the rules and principles to be followed by organizations responsible for the adoption, implementation, or imposition of anti-doping rules [20].

The obligation of every athlete, coach, medical professional, and other team members is to know and respect the valid anti-doping policies and rules adopted in accordance with the WADC. However, previous research has shown there is a low level of knowledge of athletes about doping and the need to educate them about it [20,21]. In particular, in studies that directly examined athletes' knowledge about doping problems, the highest percentage of correct answers related to questions concerning the athlete's rights and responsibilities, while the lowest percentage of correct answers related to questions about anti-doping rules and procedures [21,22]. One of the reasons for the low level of knowledge of athletes about

the legal regulations of doping is their general trust in coaches and other team members as a source of acquiring information about doping [23–25]. As a result, athletes evidently do not pay attention to acquiring proper information about anti-doping regulations, which directly puts them in danger of violating anti-doping rules.

Therefore, in improving the overall situation of that matter, it would be important to clearly identify the possible association between knowledge of the legal issues related to doping in sports and doping behavior in athletes. As a first step, it is crucial to develop a reliable and valid measurement tool aimed at assessing the level of knowledge on anti-doping legal regulations in sport. This study aims to (i) evaluate the reliability and gender-specific discriminative validity of an original questionnaire (Q-LADR), in evaluating the knowledge of legal anti-doping regulations (LADR knowledge) and (ii) to examine the gender-specific associations between LADR knowledge and potential doping behavior in senior-level professional athletes from Croatia and Montenegro. We hypothesize that greater LADR knowledge is associated with a lower tendency toward doping behavior in athletes, irrespective of gender.

## 2. Methods

### 2.1. Participants and Study Design

Participants in this study were competitive athletes from Croatia and Montenegro (*n* = 457; 179 females). All participants were older than 18 years of age and involved in Olympic team sports (football/soccer, basketball, handball, and volleyball). This sample was intentionally selected for the following reasons. First, since one of the study aims was to assess the reliability and validity of the newly developed questionnaire in evaluating LADR knowledge, it was important to observe athletes who were involved in similar sports. Otherwise, the high variability of the responses could artificially increase the test–retest reliability throughout, leading to higher numerical values of the test–retest correlation. Second, despite some technical and grammatical differences, the same language is spoken in both countries, which was one of the most important prerequisites for the effective application of the measurement tool. Third, we had to overview the "real" LADR knowledge, and for this reason, the study should be conducted within a relatively short time frame (to avoid consultations and discussion on the tested issues in sport society). Team sports are the most popular sports in the study countries, which allowed us to obtain an appropriate sample of participants in a relatively short period of time. This study was conducted in direct cooperation with the national sport authorities (i.e., committees and federations) in both countries. At the very beginning, sport authorities were contacted and they provided positive feedback about the investigation. Accordingly, the suggestion was sent by them to sport teams and athletes to participate in being contacted by the authors of the study. The authors directly contacted several teams from their region and arranged the testing. Testing comprised, altogether, 24 sports teams (9 female and 15 male teams), including 10 football/soccer, 6 basketball, 4 handball, and 4 volleyball teams.

For the purpose of the evaluation of the test–retest reliability, the subsample of participants (*n* = 87) was tested twice in a time frame of 5–7 days, depending on the possibility of gathering athletes for testing. The results were used for the evaluation of the questionnaire's reliability (please see later sections for the statistical analyses used), which was conducted immediately after retesting. The remaining sample (*n* = 370) was tested in the 2nd period (7–14 days after retesting the 87 participants, who were observed throughout the testing and retesting phases). In the remaining analysis, the data of all tested athletes were used.

### 2.2. Variables and Testing

The variables observed in this study were sociodemographic factors, sport factors, doping factors, and LADR knowledge. All variables were collected by testing participants through a specifically designed digital platform located on a server belonging to the university of the corresponding author. The participants completed the digital form of the applied questionnaires using their smartphones (the investigator provided smartphones for

those who eventually did not have one). Testing was anonymous, and all participants were informed that they could refuse to participate, and/or could leave some questions or the entire questionnaire unanswered. However, in order to allow the tracking of the responses, the participants who were tested using the test and retest procedure used the self-selected code for identification purposes. The response rate was higher than 98%, and the results of the participants who responded to all the questions were used for the analyses conducted in this study (see proceeding sections for further exclusion criteria)

The sociodemographic variables included age (in years) and gender (male–female–not applicable). In this study, we observed results for only those participants who defined their gender as male or female; therefore, one participant was not included in further analyses (based on responding with "none"). Sport factors included participating sport (football–basketball–handball–volleyball), age of initiation in sport (later used to determine "experience in sport" by subtracting the participant's age), and the highest sport achievement (local–national–international levels). Doping factors included variables asking participants about their personal opinion on doping presence in their sport (not present–doping is rare–doping is common–doping is regular in my sport), opinion about penalties for offenders of anti-doping regulations (I do not think they should be punished–financial penalty–first-time milder punishment and then exclusion from sport/competition for a couple of seasons–exclusion from sport for a couple of seasons–lifelong exclusion from sport), number of anti-doping tests in which the athlete was involved (none–one–two to three–more than three), and potential doping behavior (I will not consider doping–I do not know/not sure–I will consider doping if there are no side effects–I will consider doping). Later, the results for potential doping behavior were categorized as "negative attitude toward doping behavior" (first response), "neutral attitude" (second response), and "positive attitude" (last two responses). All sociodemographic, sport, and doping variables were tested by previously designed and validated questionnaires, and the details of the psychometric properties are available elsewhere [26].

The Q-LADR was developed through several phases. In the first phase, a panel of six experts was gathered and consulted in order to obtain a clear idea of the topics they considered important to be included in the measurement tool we intended to develop. The panel included two experts in legal issues in sport, two sport coaches with substantial experience in anti-doping protocols, and two professional athletes who were personally involved in anti-doping procedures. After the preliminary consultations, the panel agreed that the intended measurement tool should include questions on: (i) the athletes' obligations and rights with regard to anti-doping rules, and (ii) legal procedures for anti-doping testing. In the subsequent phase, the authors of the study proposed 20 questions on targeted topics (10 questions for each topic). The experts involved in the panel were asked to rank the questions in each topic from the most to the least suitable, considering (i) comprehensibility of the question, (ii) importance of the question for overall anti-doping regulations, and (iii) applicability of the question to both genders (i.e., some legal issues are exclusively characteristic for males/females while we intended to develop questionnaire that will be equally applicable to both male and female athletes). This procedure allowed us to construct a questionnaire in which the content was validated to ensure it was appropriate (e.g., have proper content validity).

The initial/longer version of Q-LADR consisted of 20 "statements", and the participants had to indicate whether the statement was correct or not. The response of "not sure" was also provided. Each correct response was scored as "1", and "0" otherwise, and the result was expressed as the sum of scores for each participant. After checking the reliability and factor validity of the questionnaire (please see proceeding sections for details of the analyses and results), the total number of questions was reduced to 10. Consequently, the final results ranged from "0" (e.g., if participant did not correctly respond to any of the questions) to "10" (all correct answers).

*2.3. Statistics*

In the first phase, we checked the reliability and validity of the Q-LADR. For the purpose of evaluating the reliability, Cohen's kappa coefficient (kappa) and the percentage of equally responded queries were calculated. These analyses were conducted on the results of those participants who were tested through a test–retest procedure. Kappa values were interpreted as slight = 0.00–0.20, fair = 0.21–0.40, moderate = 0.41–0.60, substantial = 0.61–0.80, and almost perfect = 0.81–1.00, and $p0 \geq 80\%$ was considered as acceptable. A percentage of equally responded queries of >80% was considered as acceptable.

Using the Q-LADR results of the total sample, the factor analysis using varimax rotation and Gutman Kaiser criterion of extraction was applied to check the factorial validity of the Q-LADR. This analysis identified the latent dimensions of the questionnaire and the questions related to each latent dimension. Consequently, it allowed us to reduce the number of questions for each latent dimension. After reducing the number of questions and calculating the final result of the Q-LADR for each participant, the distribution normality was checked using the Kolmogorov–Smirnov test.

The differences between genders in the Q-LADR were calculated according to the *t*-test for independent samples. One-way analysis of variance was used to evidence the differences among sports in the Q-LADR.

After checking the normality of the distributions, Spearman's rank order correlation was calculated to establish the associations between variables. Finally, multinomial logistic regression analysis was conducted to evidence the association between all variables, including Q-LADR, and criterion, namely, categorized scores of athletes' potential doping behavior (negative–neutral–positive attitudes; please see Variables subsection for details). The correlation analyses were gender-stratified.

Statistica version 13.5 (Tibco Inc., Palo Alto, CA, USA) was used for all analyses, and a *p*-level of 95% was applied.

## 3. Results

The kappa coefficient of 0.65 indicated the Q-LADR as substantially reliable. The average percentage of the equally responded queries of 84% additionally confirmed the satisfactory consistency of the measurement across the test and retest procedure. However, it appears that the consistency of the responses differed across different questions/statements. More precisely, a relatively lower consistency was observed for questions/statements No6, No13, and No16 (70–75%), while the greatest consistency was observed for statements No14 and No20 (>96% of the equally responded queries).

Factor analysis with consecutive varimax rotation evidenced three independent latent dimensions, with the first two being easily interpretable. The first latent dimension (F1) was mostly saturated with high correlations of questions/statements about the anti-doping testing protocol (No10, No12; No17, No18, and No19), where athletes were asked about procedural details and the specifics of anti-doping testing legislatives. The questions/statements concerning the athletes' rights and obligations considering WADC were mostly correlated with second significant latent dimension (F2). Namely, the highest correlations with F2 were identified for questions No2, No5, No6, No9, and No20. The third latent dimension (F3) correlated with several questions, but the structure was not clear and therefore not interpreted. From the factor analysis results, it is evident that certain questions/statements are not strongly correlated with any of the extracted factors. For example, question No1 was equally correlated with F1 and F2, and this is also evident for some other queries (i.e., No4). As a result, the set of questions finally included in the Q-LADR consisted of 10 questions; 5 items that were highly correlated with F1 (No10, No12; No17, No18, and No19) and 5 that had the highest correlations with F2 (No2, No5, No6, No9, and No20) (Table 1).

**Table 1.** Factor analysis results for original (longer) version of the Q-LADR.

| | F1 | F2 | F3 |
|---|---|---|---|
| The list of prohibited substances is published by WADA (World Anti-Doping Agency) at least once a year. [P] | 0.41 | 0.32 | 0.11 |
| Within one year, an athlete can be tested an unlimited number of times. [A] | 0.31 | 0.69 | 0.23 |
| The burden of proving a violation of anti-doping rules rests with the athlete. [A] | 0.37 | 0.22 | 0.11 |
| An athlete who is in the group designated for testing can fail to submit their location data 3 times in a period of 12 months without sanction. [A] | 0.41 | 0.32 | 0.06 |
| The athlete who is in the group designated for testing is obliged to send information about the location for each day in the period for the next 3 months. [A] | 0.20 | 0.71 | 0.03 |
| An athlete can refuse to provide a sample to the organization responsible for testing if he believes that the organization that will test him has political or other prejudices about him (for example, organizations from countries that have political conflicts with the athlete's country). [A] | −0.03 | 0.61 | 0.27 |
| The athlete must request a therapeutic exemption if he is forced to take a drug that is on the List of Prohibited Substances. [A] | 0.12 | 0.12 | 0.24 |
| A therapeutic exemption can be requested retroactively, but before the testing is conducted. [A] | 0.09 | 0.18 | 0.12 |
| The doping control officer must inform the athlete several hours before he will come to test him. [P] | 0.04 | 0.78 | 0.15 |
| The doping control officer can take a urine and blood sample from the athlete. [P] | 0.79 | 0.23 | 0.15 |
| When giving a urine sample, the athlete chooses the container to collect the sample himself. [P] | 0.15 | 0.31 | 0.09 |
| Samples can be stored and reanalyzed over a period of 10 years. [P] | 0.71 | 0.21 | 0.22 |
| During the period of ineligibility, the athlete is still subject to testing. [P] | 0.12 | 0.01 | 0.34 |
| If the A sample is positive, the athlete has the right to request the B sample's analysis. [A] | 0.37 | 0.08 | 0.03 |
| A temporary suspension is mandatory if the A sample is positive. [P] | 0.67 | 0.10 | 0.50 |
| The athlete may be present during the opening and analysis of the B sample. [A] | 0.58 | 0.02 | 0.44 |
| If the analysis of sample B does not confirm the analysis of sample A, the test will be considered negative. [P] | 0.70 | 0.05 | 0.34 |
| An appeal can be filed against a decision on a violation of anti-doping rules. [P] | 0.73 | −0.12 | 0.16 |
| If an athlete is found guilty of violating anti-doping rules, the fact will be made public. [P] | 0.78 | 0.22 | 0.04 |
| An athlete who voluntarily admits to having violated an anti-doping rule before receiving notification of the violation may have his or her suspension period shortened. [A] | 0.45 | 0.73 | −0.15 |
| Explained Variance | 4.57 | 3.06 | 1.51 |
| Proportion Total | 0.23 | 0.15 | 0.08 |

Legend: [A] specify the questions initially considered to be related to athletes' obligations/rights in anti-doping testing, [P] specify the questions initially considered to be related to anti-doping protocol, F—factor structure (correlations with variables).

The *t*-test for independent samples identified significant differences in Q-LADR between genders (Figure 1A), with higher test scores for males (*t*-test = 9.55, *p* < 0.001). Differences among sports in Q-LADR (Figure 1B) were not significant (F-test = 0.72, *p* = 0.73).

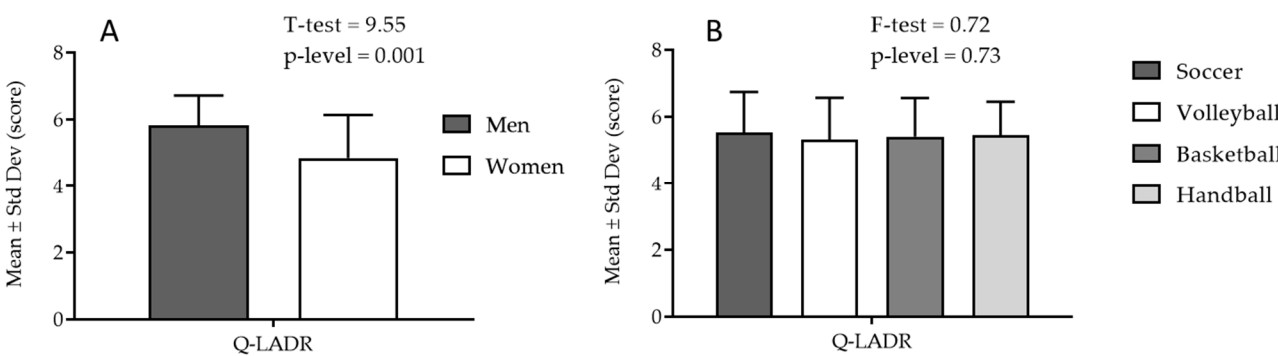

**Figure 1.** Differences in Q-LADR between genders (**A**) and among sports (**B**).

The correlations among the study variables are presented in Table 2. Apart from logical correlations (i.e., correlations between athletes' age and sport experience, sport success, and age), some interesting associations noting the specific correlations between study variables and Q-LADR scores were presented. For men, the older athletes and those who achieved

greater sport success had higher Q-LADR scores. Furthermore, better sport success was correlated with personal opinions about the necessity for more rigid penalties for doping offenders in both men and women. Moreover, Q-LADR was correlated with the number of doping tests in both men and women, with better Q-LADR scores for those athletes who frequently participated in doping tests. For women, sport success and experience were significantly positively correlated with Q-LADR.

**Table 2.** Correlations between study variables (* denotes significance of $p < 0.005$).

|  | Age | Sport Experience | Sport Achievement | Doping Testing | Doping Presence | Doping Penalties |
|---|---|---|---|---|---|---|
| MEN (*n* = 278) |  |  |  |  |  |  |
| Sport experience | 0.55 * |  |  |  |  |  |
| Sport achievement | 0.51 * | 0.32 * |  |  |  |  |
| Doping testing | 0.19 * | 0.08 | 0.08 |  |  |  |
| Doping presence | 0.16 * | 0.14 * | 0.05 | 0.21 * |  |  |
| Doping penalties | 0.24 * | −0.11 | 0.42 * | −0.15 * | −0.11 |  |
| Q-LADR | 0.38 * | 0.16 * | 0.38 * | 0.41 * | 0.35 * | −0.11 |
| WOMEN (*n* = 179) |  |  |  |  |  |  |
| Sport experience | 0.61 * |  |  |  |  |  |
| Sport achievement | 0.25 * | 0.23 * |  |  |  |  |
| Doping testing | 0.29 * | 0.10 | 0.16 * |  |  |  |
| Doping presence | −0.07 | −0.24 * | −0.08 | −0.05 |  |  |
| Doping penalties | 0.06 | 0.05 | 0.50 * | −0.02 | 0.05 |  |
| Q-LADR | 0.11 | 0.21 * | 0.61 * | 0.49 * | 0.11 | 0.05 |

Among men, a higher likelihood for neutral (OR = 1.62, 95%CI: 1.21–2.03) and positive attitudes toward PDB (OR = 1.71, 95%CI: 1.11–2.22) was observed in the athletes who perceived their sport as being contaminated by doping. Additionally, men with positive PDBs were less likely to achieve high competitive results in sport (OR = 0.56, 95%CI: 0.35–1.90). Finally, a greater likelihood for positive PDB was evidenced in those who achieved lower scores for the Q-LADR (OR = 0.75, 95%CI: 0.5–0.96) (Figure 2).

The women who tended toward rigid penalties for doping offenders were less prone to neutral (OR = 0.81, 95%CI: 0.70–0.92) and positive (OR = 0.90, 95%CI: 0.80–0.98) PDBs. Better competitive results are correlated with a lower possibility for positive PDB (OR = 79, 95%CI: 0.60–0.92). In women, lower Q-LADR scores were correlated with neutral (OR = 0.88, 95%CI: 0.75–0.99) and positive (OR = 0.79, 95%CI: 0.60–0.92) PDBs (Figure 3).

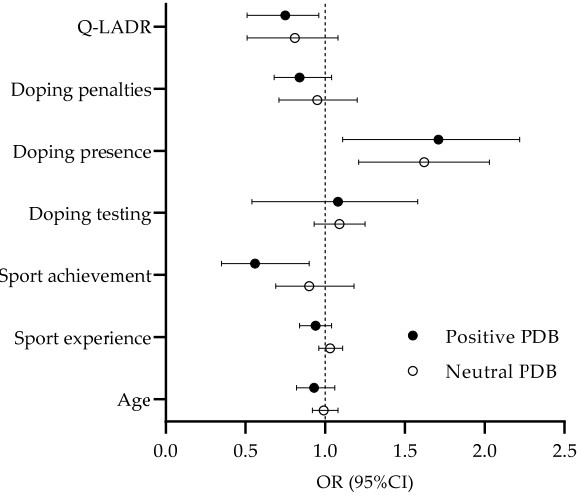

**Figure 2.** Multinomial logistic regression results for the criterion of potential doping behavior (PDB) in men, with negative PDB as reference value.

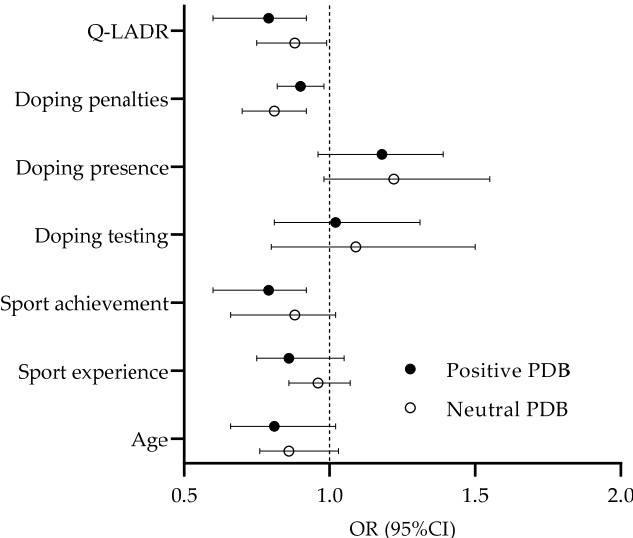

**Figure 3.** Multinomial logistic regression results for the criterion of potential doping behavior (PDB) in women, with negative PDB as reference value.

## 4. Discussion

This study aimed to assess the applicability of the newly develop questionnaire, Q-LADR, in evaluating the knowledge on legal issues of the anti-doping policy (LADR knowledge) in professional-team-sport athletes, and to investigate its gender-specific validity with regard to potential doping behavior. Apart from evidencing the sufficient reliability of the newly designed measurement tool, the results reveal several important observations. First, Q-LADR differentiated male and female athletes involved in team sports, with men demonstrating better knowledge. Second, Q-LADR was observed to be correlated with age in men, and the number of doping tests and sport success in men and women. Finally, Q-LADR was correlated with doping attitudes, with a generally lower likelihood for doping behavior in athletes who achieved higher Q-LADR scores. Therefore, our initial study hypothesis was confirmed.

### 4.1. Gender Differences in Q-LADR

One of the main ideas of this study was that doping problems and factors associated with doping problems in sport should be observed using a gender-specific approach. The differences in Q-LADR between men and women, where men achieved higher scores for Q-LADR, supported the use of such an approach. As the authors of this study, we are long-time professionals in sport studies and suggest that the a background of higher scores in men than women should be found in several practical reasons.

First, doping is more prevalent in male competitive sports, which is evident by simply observing the annual WADA reports [27]. Meanwhile, even in noncompetitive sports (i.e., recreational sport), doping is more prevalent in males than in females [28,29]. Altogether, this raises the interest in doping among men more than women, independently of whether they are involved in a competitive sport system or not. This altogether results in men being relatively better informed about doping than women.

Second, men are more involved in sport and persist in competitive sport for a longer period of time than women [30]. Indeed, gender is an indefinite position of inequality in sport [31]. Although this issue is partially related to the fact that sport is organized around the "discourses of hegemonic masculinity" [32], we cannot deny that there are also some "objective" reasons for the generally longer sport involvement of men in comparison to women. These "objective" reasons are as much biological (e.g., menstrual cycle is a significant burden for sport participation) as "ecological" (e.g., motherhood represents a much greater obligation than fatherhood) [33–35]. However, irrespective of the background, the average age of male athletes is regularly greater than the average age of female athletes

involved in the same sports, which altogether leads to the greater chance that "male society" in sport would be properly informed in any sports issues (including doping) than their female peers.

As a certain support system to the previous discussion, a short overview of the previous studies is important. In brief, studies have already examined gender differences in the knowledge of doping with interesting results. For example, in a study conducted on junior swimmers (<18 years of age), the authors evidenced an equal level of knowledge concerning doping issues in boys and girls [21]. Furthermore, a study conducted with team-sports players did not report differences between genders concerning doping knowledge [13]. However, in both studies, there was no difference in age between the male and female participants. Meanwhile, a study on senior tennis players where male players were older than the females evidenced a better knowledge of doping issues in males [20], clearly emphasizing the importance of experience in sport as a factor of influence on various doping-related factors. This issue was additionally discussed in the forthcoming text.

### 4.2. Correlates of Q-LADR

The knowledge of LADR was correlated with age, which actually additionally supports our previous discussion about the probable background of differences in Q-LADR between genders. It was clear that the overall time in which one was involved in sport reflected the opportunity to be informed about some specific sport issue. Studies, to date, have already confirmed the correlation between knowledge of doping with age in a gender-stratified study of tennis professionals, but also in a non-gender-stratified swimming study [20,21]. Therefore, we may say that our results are generally supportive of the previous findings and even previously offered explanations. In brief, older athletes have simply had more chances to familiarize themselves with the issue of interest (here being the legal regulations on doping), either from personal experience (i.e., more anti-doping testing) and/or by direct formal/non-formal learning.

That this correlation between Q-LADR and age was not established in women in our study (note that in a previously discussed study on tennis, the correlations between doping knowledge and age were established both for men and women) can be explained by the differences in the studied samples and sports characteristics [20]. In short, team sports are more "gender separated" than in tennis. In tennis, men's and women's tournaments are frequently organized simultaneously, while even training camps are organized for men and women at the same time [36]. At the same time, this is not the case for team sports, where men and women train and compete separately (with the exemption of the Olympics). This reduces the overall chance for information exchange between male and female colleagues, resulting in a significant association between age and Q-LADR, only in male-team-sports athletes.

Although we could not find a study where the problem of "exchange of information on doping" and its influence on the knowledge of doping was directly examined, the studies conducted so far indirectly support our previous discussion. Namely, investigations that focused on sources of information on doping problems in athletes regularly evidenced colleagues (peers) and coaches as the most important sources of knowledge on doping issues. This was repeatedly confirmed in various sports and athletes of a different competitive status [24,37–39]. Therefore, it was reasonable to conclude that the similar "mechanism" of influence exists, even for knowledge of LADR.

The correlation of Q-LADR with (i) the number of doping tests and (ii) sport achievements was significant for both men and women. These two associations were actually "confounding" (the chances for being tested for doping increased with sport success) and have already been presented in other studies [13], and we discussed this accordingly. As evident, most of the questions included in the final version of the Q-LADR went beyond "common knowledge" and directly incorporated information about strict legal regulations that have to be specifically learned. We may witness that in both of the countries we observed in this study, the information related to Q-LADR was rarely, if ever, systemati-

cally taught throughout the sport systems (by sport authorities, clubs, federations, etc.). Therefore, the "only" opportunity for athletes to increase their knowledge of these issues was to participate in anti-doping tests and directly acquire the knowledge in the real world. For example, if an athlete participated in anti-doping tests, they undoubtedly learned that "WADA officials must legitimize themselves before testing" and/or that "the doping control officer does not have to inform the athlete several hours before he will come to test him". Altogether, this explained the established correlations, but at the same time highlighted the vulnerability of younger athletes (both men and women) in violating anti-doping rules and regulations.

### 4.3. Q-LADR as a Factor Associated with Potential Doping Behavior

As noted in the Introduction, investigations have frequently examined the correlations of doping behavior/doping attitudes, but the knowledge on doping issues as a correlate of doping behavior has hardly been investigated [11,24,25]. There are several probable reasons for such a lack of studies on this problem. First, the world governing body for anti-doping in sport, WADA, as developed an accessible and applicable questionnaire ("Play true quiz"), which is freely available on the WADA website; this "measurement tool" was, however, evidently developed with the main intention of "educating" and not "testing", since the correct answers are indefinitely predictable and intuitive [40]. Therefore, the results obtained do not represent the examinee's knowledge, but rather the level of their intuition. Second, when investigators intend to develop an objective testing tool, they should respect specific sport/geographic/cultural issues and construct the questions accordingly. Therefore, if there is an intention to apply it in some other environment, the questionnaires should in most cases be adapted. Finally, the doping questionnaires developed to date, mostly focus on the health issues of doping (or at least comprised a significant proportion of questions related to health issues), and globally, there is a lack of precise information on the problem examined in the Q-LADR.

Despite some differences in the statistical significance, Q-LADR was correlated to the doping attitudes of both men and women, with less negative attitudes toward doping in athletes who achieved lower Q-LADR scores (have poorer knowledge of anti-doping legal regulations). This finding confirms the questionnaire's validity and supports our initial study hypothesis. However, for a more objective overview of the (predictive) validity of the questionnaire, additional correlations of doping attitudes (criterion) should be taken into account and discussed in parallel. A relatively consistent finding of wider knowledge of the legal regulations of doping among athletes with a negative tendency toward doping is not as straightforward as it may appear at first. Specifically, we must pay attention to our finding that greater sports achievements are also correlated with a lower tendency for doping, which is a known issue [13]. Therefore, it is possible that a certain degree of the correlation between Q-LADR and doping attitudes could be explained by confounding effects (e.g., intercorrelation between sport success and Q-LADR).

One can therefore argue that the Q-LADR score is a consequence of sport success (i.e., experience in sport and number of anti-doping tests an athlete was involved in) simply by confounding the statistical effect [41]. If this was the case, the Q-LADR itself would not represent a great significance with regard to doping-prevention efforts. However, it is important to note that neither the experience in sport or the number of doping tests were significantly related to doping attitudes in the multinomial logistic results. Therefore, the correlation of the Q-LADR and doping attitudes should be observed independently of the experience the athletes obtained throughout their time involved in sport and the number of anti-doping tests. From the perspective of anti-doping-prevention efforts, this is the most important finding. Putting the results together, we may conclude that those athletes who achieve better sport success and those who have higher Q-LADR scores are less prone to doping. However, sport success cannot be achieved overnight, and due to all the inevitable determinants (i.e., one's talent, conditions, coaching, finances, and sport luck) is only achievable for a small proportion of athletes. Meanwhile, a sufficient

level of knowledge of the legal issues of doping is attainable for all athletes and depends mostly (only?!) on: (i) an awareness of the necessity and (ii) the proper organization of the education.

### 4.4. Limitations and Strengths of the Study

The main limitation of this study was its cross-sectional nature. Therefore, we were unable to discuss the causality between the variables. However, this study primarily intended to explore the eventual importance of the legal issues of anti-doping regulations in sport and provide a base for future, more profound investigations of the problem. Second, we only observed athletes involved in team sports in two countries. Although we did this intentionally (please see Introduction and Methods Sections for details), there is no doubt that the presented findings are consequently generalizable only to similar samples. Moreover, the sample of variables did not include some variables that would certainly allow for a more comprehensive overview and more detailed discussion (i.e., familial variables, socioeconomic status, and educational level). However, we relied on the honesty of the participants, with testing conducted in small groups, and we tried to avoid any variables that could compromise the anonymity of the tested athletes.

This was one of the first studies to directly examine the knowledge of the legal issues of anti-doping regulations in sport while developing an original measurement tool for such a purpose. In addition, the fact that we investigated professional athletes involved in the most popular types of sport in southeastern Europe (e.g., team sports) was another important strength of the investigation. Finally, the analyses of the factors related to doping behavior were known to be relevant in developing targeted preventive campaigns against doping in sport; therefore, we hope that our study contributes to the body of knowledge on this important topic.

### 5. Conclusions

Our results confirm that female athletes are placed at a substantially higher risk of violating anti-doping rules because of being inadequately informed about anti-doping legal regulations they have to obey as competitive athletes. From the position of sport and legal authorities, this is irresponsible (at least), but we would rather call it discriminatory.

The study confirmed the reliability and applicability of the Q-LADR in senior-level professional-team-sports players from Croatia and Montenegro. However, it must be mentioned that the developed measurement tool was specifically designed for our countries, and it is possible that some questions will not be absolutely applicable to other countries. Therefore, in applying the Q-LADR in other countries, special attention should be placed on checking the proposed questions here and eventual modifications according to the current national laws.

Athletes who achieved better results in the Q-LADR were less prone to doping and it seems that the knowledge of anti-doping regulations could be effectively used in anti-doping efforts. Namely, we may theorize that the improvement of knowledge on legal issues about anti-doping would decrease the positive tendencies toward doping behavior among athletes. However, for a more elaborate conclusion, further intervention studies aimed at the improvement of knowledge regarding the legal issues of anti-doping regulations are needed.

**Author Contributions:** Data curation, D.V.S., A.S., D.S., M.J. and J.R.; formal analysis, D.V.S.; methodology, M.J.; project administration, D.V.S. and M.J.; resources, D.V.S. and J.R.; software, J.R.; writing—original draft, D.S. All authors have read and agreed to the published version of the manuscript.

**Funding:** This research received no external funding.

**Institutional Review Board Statement:** The study was conducted in accordance with the Declaration of Helsinki, and approved by the Institutional Ethics Committee of University of Split, Faculty of Kinesiology (protocol code 2181-205-02-05-14-004; 17 June 2014).

**Informed Consent Statement:** Informed consent was obtained from all subjects involved in the study.

**Data Availability Statement:** Data will be provided to all interested parties upon reasonable request.

**Conflicts of Interest:** The authors declare no conflict of interest.

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
