# Peer review of "Knowledge of the Legal Issues of Anti-Doping Regulations: Examining the Gender-Specific Validity of the Novel Measurement Tool Used for Professional Athletes"

_sustainability, doi:10.3390/su141912883_

Round 1

Reviewer 1 Report

The main aim of this study was to assess the reliability and gender-specific validity of an original questionnaire (Q-LADR) in evaluating knowledge on legal anti-doping regulations and to examine the gender-specific associations between Q-LADR and potential doping behavior (PDB) in senior-level professional athletes. Regarding the authors, I would like to congratulate and thank them for their effort and motivation involved in this research study. The presentation of the research is well documented, with a scientific basis and respects the latest standards regarding the highest level scientific publications. The methodology was chosen correctly. The conclusions support and result from the research and open new directions for future research. The submitted work is interesting, but in my opinion the weakest part is the discussion, referring to a small number of references and not very substantive. Please strengthen it by adding similar studies and comparing them.

Please also add the authors’ contribution, as failure to disclose this at the first stage of the review may cause doubts as to whether all the indicated authors actually contributed to the research and to the writing of the manuscript.

Although the authors have declared the data to be made available to all interested parties upon reasonable request, in my opinion this type of study should be fully public and I suggest the authors add the results in supplementary material. This will increase the transparency of the study.

I keep my fingers crossed for the final success of the publication, which in my opinion is really good.

Author Response

The main aim of this study was to assess the reliability and gender-specific validity of an original questionnaire (Q-LADR) in evaluating knowledge on legal anti-doping regulations and to examine the gender-specific associations between Q-LADR and potential doping behavior (PDB) in senior-level professional athletes. Regarding the authors, I would like to congratulate and thank them for their effort and motivation involved in this research study. The presentation of the research is well documented, with a scientific basis and respects the latest standards regarding the highest level scientific publications. The methodology was chosen correctly. The conclusions support and result from the research and open new directions for future research. The submitted work is interesting, but in my opinion the weakest part is the discussion, referring to a small number of references and not very substantive. Please strengthen it by adding similar studies and comparing them.

RESPONSE: Thank you for your support and for recognizing the potential of our work. We tried to follow your suggestions and comments and amended the manuscript accordingly. With regard to your comment on Discussion, we must say that in this version of the manuscript we tried to follow your main suggestion (i.e. small number of references) and used additional references, specifically:

  1. Eime, R.; Charity, M.; Harvey, J.; Westerbeek, H. Five-year changes in community-level sport participation, and the role of gender strategies. Frontiers in sports and active living 2021, 3.
  2. Spaaij, R.; Farquharson, K.; Marjoribanks, T. Sport and social inequalities. Sociology Compass 2015, 9, 400-411.
  3. Leberman, S.; Palmer, F. Motherhood, Sport Leadership, and Domain Theory: Experiences From New Zealand. Journal of Sport Management 2009, 23.
  4. Constantini, N.W.; Dubnov, G.; Lebrun, C.M. The menstrual cycle and sport performance. Clinics in sports medicine 2005, 24, e51-e82.
  5. Brown, N.; Knight, C.J.; Forrest, L.J. Elite female athletes’ experiences and perceptions of the menstrual cycle on training and sport performance. Scandinavian Journal of Medicine & Science in Sports 2021, 31, 52-69.
  6. Żurek, P.; Lipińska, P.; Antosiewicz, J.; Durzynska, A.; Zieliński, J.; Kusy, K.; Ziemann, E. Planned Physical Workload in Young Tennis Players Induces Changes in Iron Indicator Levels but Does Not Cause Overreaching. Int J Environ Res Public Health 2022, 19, doi:10.3390/ijerph19063486.
  7. Muwonge, H.; Zavuga, R.; Kabenge, P.A. Doping knowledge, attitudes, and practices of Ugandan athletes': a cross-sectional study. Subst Abuse Treat Prev Policy 2015, 10, 37, doi:10.1186/s13011-015-0033-2.
  8. Juma, B.O.; Woolf, J.; Bloodworth, A. The challenges of anti-doping education implementation in Kenya: Perspectives from athletes and anti-doping educators. Performance Enhancement & Health 2022, 10, 100228.
  9. Loraschi, A.; Galli, N.; Cosentino, M. Dietary supplement and drug use and doping knowledge and attitudes in Italian young elite cyclists. Clinical Journal of Sport Medicine 2014, 24, 238-244.

Also, discussion is significantly rewritten in some parts as you suggested. For example, first subheading of the discussion includes the paragraph where we overviewed previous studies where authors discussed differences between genders in doping knowledge. Text reads: “As a certain support to the previous discussion a short overview of the previous stud-ies in important. In brief, studies already examined gender-differences in knowledge on doping with interesting results. For example, in a study done with junior swimmers (< 18 years of age), authors evidenced equal level of knowledge on doping-issues in boys and girls [23]. Further, study done with team sport players didn’t report differences between genders in doping-knowledge [13]. However, in both studies there was no difference in age between male and female participants. Meanwhile, study on senior tennis players where male players were older than females evidenced better knowledge on doping issues in males [22], clearly emphasizing the importance of experience in sport as a factor of influence on various doping-related factors. This issue is additionally discussed in the forthcoming text.” (please see last paragraph under subheading 4.1. Gender differences in Q-LADR). Also, in the second subheading of the discussion we included text on “sources of information on doping”. Text reads: “Although we couldn’t find study where the problem of “exchange of information on doping” and its influence on knowledge on doping was directly examined, studies done so far indirectly support our previous discussion. Namely, investigations that were focused on sources of information on doping-problems in athletes regularly evidenced colleagues (peers), and coaches as the most important as the most important source of knowledge on doping issues. This is repeatedly confirmed in various sports and athletes of different competitive status [26,39-41]. Therefore, it is reasonable to conclude that the similar “mechanism” of influence exists even for knowledge on LADR.” (please see 3rd paragraph of the subheading 4.2. Correlates of Q-LADR), etc.

However, we must note that second reviewer raised concern about “overload of the text” so we paid attention on it at the same time. Therefore, the overall size of the discussion (i.e. with regard to number of words) is not substantially changed (in the original version 1693 words, and now approximately 2000 words).

Please also add the authors’ contribution, as failure to disclose this at the first stage of the review may cause doubts as to whether all the indicated authors actually contributed to the research and to the writing of the manuscript.

RESPONSE: Thank you, the details are added and text now reads: “Data curation, Draginja Vuksanovic Stankovic, Antonela Sinkovic, Damir Sekulic, Mario Jelicic and Jelena Rodek; Formal analysis, Draginja Vuksanovic Stankovic; Methodology, Mario Jelicic; Project administration, Draginja Vuksanovic Stankovic and Mario Jelicic; Resources, Draginja Vuksanovic Stankovic and Jelena Rodek; Software, Jelena Rodek; Writing – original draft, Damir Sekulic.” (please see subsection on Authors Contribution)

Although the authors have declared the data to be made available to all interested parties upon reasonable request, in my opinion this type of study should be fully public and I suggest the authors add the results in supplementary material. This will increase the transparency of the study.

RESPONSE: Indeed, we agree the availability of the data will definitively increase the transparency. However, the study is done under jurisdiction of the National Sport Authorities (Federations and Olympic Committees – please see Participants under Methods subsection and Responses to 2nd Reviewer). As our partners actually retain the rights on data we are not able to provide data publicly. Meanwhile, we assure you that any individual request will be adopted.

I keep my fingers crossed for the final success of the publication, which in my opinion is really good.

RESPONSE: Thank you!

Staying at your disposal!

Reviewer 2 Report

Thank you for the possibility to review that paper. I have few comments:

1. The title should be more informative. After reading it, you do not expect that the study was to evaluate the questionnaire.

2. I would also suggest to consider some other key words more related with the topic.

3. Introduction: I would suggest to shorten the Introduction a little bit (line 78-110). It should be more focused on the topic, so the lack of tools to measure the knolwedge of doping rules and only highlight the problem of knowledge about doping. 

4. Methods: How did you find the participants? Voluntary? By on-line message? They were from the whole country so how were you able to get to all of them.

5. Conclusions should be shorten giving only the short summary of the study, the practical application and that's all.

6. Some references are only links. Can you give some name?

Generally, the manuscript is weel-written, however some time I think that the text is overloaded.

Author Response

Thank you for the possibility to review that paper. I have few comments:

  1. The title should be more informative. After reading it, you do not expect that the study was to evaluate the questionnaire.

RESPONSE: Thank you; the title is changed accordingly and now reads: Knowledge on legal issues of anti-doping regulations: examining gender-specific validity of the novel measurement tool in professional athletes

  1. I would also suggest to consider some other key words more related with the topic.

RESPONSE: Key words are changed (questionnaire; validity; legislative; sport law; integrity). Thank you.

  1. Introduction: I would suggest to shorten the Introduction a little bit (line 78-110). It should be more focused on the topic, so the lack of tools to measure the knolwedge of doping rules and only highlight the problem of knowledge about doping.

RESPONSE: We must agree that the Introduction was relatively “densed”; but we must also say that we did it intentionally. Specifically, the problem of legal issues in anti-doping efforts is generally understudied and we tried to cover all aspects of this problem. However, and as you suggested that some parts of the text are not necessary, we reduced the Introduction following your suggestion. Specifically, the part of the text you are mentioning in your comment (originally more than 400 words) is now reduced to approximately 200 words (please see highlighted text in the Introduction starting from approximately line 77). Thank you!

  1. Methods: How did you find the participants? Voluntary? By on-line message? They were from the whole country so how were you able to get to all of them.

RESPONSE: We definitively missed to report the sampling procedure. However, we must mention that both countries are relatively small (Croatia less than 4 million, Montenegro < 700.000 residents), so the sampling was not very difficult. In this version of the manuscript sampling was explained in more details and now reads: “National sport authorities (i.e. Committees, Federations) in both countries were contacted before the study and they provided positive feedback about the investigation. Accordingly, the suggestion was sent by them to sport teams and athletes to participated upon being contacted by authors. While authors of the study were/are personally involved in team sports, they contacted several teams from their region and arranged the testing. Testing comprised altogether 24 sport teams (9 female, and 15 male teams), including 10 football/soccer teams, 6 basketball teams, 4 handball teams and 4 volleyball teams.” (please see participants subsection).

  1. Conclusions should be shorten giving only the short summary of the study, the practical application and that's all.

RESPONSE: Indeed, the Conclusion was to long. We reduced it significantly (from 340 words to 190 words).

  1. Some references are only links. Can you give some name?

RESPONSE: All references are now named. Specifically:

  1. Croatian Law on Sports; https://www.zakon.hr/z/300/Zakon-o-sportu.
  2. Montenegrin Law on Sports: https://www.gov.me/dokumenta/8bf04168-cab3-49ce-9a85-611488870d25.
  3. WADA annual report; https://www.wada-ama.org/en/resources/annual-report.

Generally, the manuscript is well-written, however some time I think that the text is overloaded.

RESPONSE: Thank you once again for your support and suggestion. WE tried to follow all your comments and believe that it improved the readability of the manuscript.

Staying at your disposal!

Round 2

Reviewer 2 Report

Thank you for you answers and corrections. The article is fine for me, now. Congratulations!